# The Use of ICTs to Support Social Participation in the Planning, Design and Maintenance of Public Spaces in Latin America

**Sergio Alvarado Vazquez *** , **Ana Mafalda Madureira** , **Frank O. Ostermann** and **Karin Pfeffer**

Faculty of Geo-Information Science and Earth Observation (ITC), University of Twente,
7514 AE Enschede, The Netherlands; m.madureira@utwente.nl (A.M.M.); f.o.ostermann@utwente.nl (F.O.O.);
k.pfeffer@utwente.nl (K.P.)
***** Correspondence: s.alvaradovazquez@utwente.nl; Tel.: +31-684664042

**Abstract:** Recent research indicates that information and communication technologies (ICTs) can support social participation in the planning, design and maintenance of public spaces (PDMPS), specifically to create comprehensive knowledge among different stakeholders. However, critics point out that the use of ICTs by planners and decision-makers often ignores the needs of local residents. For this research, we inquired how ICTs can support social participation in PDMPS. Our case study combines a literature review and 21 semi-structured interviews with government officials, non-governmental organisations, academics and architecture/urban planning consultancy companies in Mexico to understand how different stakeholders use ICTs to improve the quality of public spaces. We developed an approach that facilitates the analysis of ICT aspects related to hardware and software supporting social participation in PDMPS. The findings show that Mexico has a base of digital tools requiring technical capacities and spatial literacy at various stages of PDMPS, and ICTs are seen as an opportunity to engage with residents. However, in practice, our interviewees mentioned that ICTs are rarely implemented to support participatory processes due to high costs, a lack of political support and the insufficient technical expertise of technical staff to engage with residents using ICTs. The paper closes with recommendations and suggestions for future research on how ICTs can better support participatory processes in PDMPS.

**Keywords:** public space management; technology; social participation; Mexico; urban planning; urban design; Latin America

## 1. Introduction

Residents are the final users of public spaces but are usually not involved in planning processes to develop new public spaces or in designing how new and existing public spaces should look or be managed [1–3]. Recent research highlights the need for social participation in the planning, design and maintenance of public spaces (PDMPS) and for governmental institutions to explore how to enhance interaction and coordination with residents and to collect local spatial knowledge [4]. Local spatial knowledge (LSK) refers to the residents' expertise in and experience with the contextual conditions in a specific place. It allows local residents and other stakeholders to share information and collaborate to solve urban issues [5] and supports bottom-up planning processes [6].

Various studies claim that the use of information and communication technologies (ICTs) can provide opportunities to facilitate processes that involve social participation and to engage and leverage LSK in PDMPS [5,7,8]. However, using ICTs for participatory planning and eliciting LSK faces numerous challenges. Experts and decision-makers usually do not consider local residents to be a main source of information when ICTs are used for planning and decision making [9]. The lack of acceptance of ICTs by government institutions and other stakeholders is another challenge found among the aforementioned actors. For example, some digital tools are not user-friendly or do not provide a feeling of

secure personal privacy [10]. The ability to connect with the community's future visions through technology while realising a potential benefit for planning professional work still requires advanced development and programming to be meaningful. The use of technology in urban planning differs from what urban planners learn during their education or from their experience, as their training often lacks a focus on technology. As a result, utilising technology in their practice may be outside their comfort zone and not within their areas of expertise [11].

To enhance our understanding of ICTs' potential to support social participation in PDMPS, this study analysed two main elements of how different stakeholders use ICTs to facilitate participatory processes: (1) the main characteristics of the ICTs used in PDMPS processes and (2) the main challenges and opportunities of using these tools to support social participation in PDMPS. This research draws on a literature review of existing studies of ICT use in supporting participatory processes in the context of PDMPS. The desktop research component was complemented with 23 semi-structured interviews with government officials, academics, NGOs and private architecture/planning consultancy companies involved in PDMPS in Mexico. For the empirical grounding, we adopted an explorative qualitative approach using two case studies as part of previously conducted research on public space management focusing on two cities of the Mexico City Megalopolis: Mexico City and Puebla [4]. We developed an approach that facilitates the analysis of ICT aspects related to hardware, software and purpose that facilitate social participation in PDMPS.

This paper is divided into five sections. Section 2 discusses the role of ICTs in supporting social participation in planning processes. Section 3 briefly outlines the case study context and describes the overall approach, the literature review procedure and the methods used to administer and analyse the semi-structured interviews. Section 4 presents the findings based on a table of aspects of ICT use and the four groups of interviewed stakeholders. Section 5 discusses the key insights from the literature review and the application of ICTs in the case study. The paper concludes by presenting the scientific and social relevance of ICTs, the challenges and opportunities of using ICTs in participatory processes, and recommendations to enhance the planning, design and maintenance of public spaces within the Latin American context, with potential transferability to the international context.

## 2. The Role of ICTs in Supporting Social Participation in Planning Processes

This work adopted the concept of social participation due to its focus on the value of involving local residents in the planning process, for instance, the planning of public spaces [12]. Participation is understood here as the act of taking part in something. It is a democratic right, as anyone can be involved in a policy process [13]. Social participation has been credited with opening up planning processes to democratic scrutiny and as a tool to inform whether a project or proposal will be accepted by its future users [13,14].

Numerous studies have affirmed that ICTs facilitate participatory processes. First, online community mapping platforms help residents to document and share the living conditions in the neighbourhood, improving collaboration between grassroots communities and local governments [15]. Second, to promote inclusive design, 3D modelling software and virtual reality (VR) devices have been used as a channel of interaction with the urban design process. The findings show that 3D modelling led to higher stakeholder engagement and feedback than using 2D paper plans about a project design [16,17]. Third, digital engagement campaigns using social media and web-based mapping services have supported large-scale social participation in urban planning districts, including public spaces; this helped address the lack of information and communal awareness [18].

ICTs generate data and information that are more readily available to stakeholders and government officials with access to an internet connection and basic ITC literacy. Traditional analogue models use printed maps or physical surveys that require processing and analysis [19]. For example, geographic information systems are used through internet web-based applications to collect insights from stakeholders for the management of public

spaces or the design of green infrastructure [5,20,21]. In addition, Public Participatory Geographic Information Systems (PPGISs) were used to bridge the gap between public servants or planners and local residents, sometimes giving a voice to and empowering local residents on urban issues [22,23].

Previous studies have emphasised how ICTs can be used to better align urban projects with residents' needs and aspirations. Examples of this are the use of interactive devices to increase participatory processes in the public space [1] or the exploration of affordable ICTs based on the principles of digital democratic affordance, a concept that addresses how emergent digital tools could help to organise and represent the interests of social organisations [2]. Another approach involves operationalising different areas of knowledge and technologies to allow local residents to participate meaningfully in planning, designing or maintaining public spaces [3–5]. As long as they are not used as isolated solutions, digital platforms can become supportive tools that could enhance communication and interaction among residents and decision-makers to manage public spaces [8,24].

Nevertheless, research has also highlighted three issues that require further exploration. First, communication in a participatory process with local residents and government institutions frequently lacks the means for meaningful feedback, which impedes the translation of local resident insights into real projects [8,25,26]. Second, the use of ICTs has not yet been effectively embedded in regular processes to support PDMPS in developing countries due to legislative and regulatory gaps in using ICTs as a source of information and communication [4,27]. Third, digital inequality persists, as not everyone has the same level of access to ICTs or the same level of ICT literacy, both of which are important aspects of the problem [11,19].

## 3. Methodology

### 3.1. Overall Research Design

Our research focused on public spaces at the neighbourhood scale that are owned by the government, have flexible functions, are appropriated by a community and are privately or publicly maintained. These spaces include parks, green areas, empty land areas, squares, plazas or the popular neighbourhood corner (often threatened by urban growth in marginalised neighbourhoods) [28,29].

This study used a mixed-method approach to increase the reliability of our findings [30]. We combined a focused literature review with semi-structured interviews and compared what has already been researched in previous studies and what our interviewees highlighted. The analysis used the lenses of (1) the public space management framework for the Latin American context, specifically the communication phase [4,31], and (2) the democracy diagram for PDMPS, particularly the dimensions of authority and power [12]. In previous research, the adoption of the public space management framework and the 'democracy diagram' proved useful in analysing participation in decision making through communication with various stakeholders, as well as in understanding how public spaces can and should be maintained [4,12].

#### 3.1.1. Public Space Management Framework

Public space management (PSM) concerns the regulation of planned and existing public spaces and the provision of facilities, as well as the quality assurance, design and management of public spaces [29]. The PSM framework identifies core issues in the planning, design and maintenance phases of public spaces. Understanding the context of public spaces allows us to understand how public spaces are idealised, the aspirations for an ideal public space, and how they function as they change over time and through different geographical contexts [2]. Defining what a public space is allows us to identify the aspirations, functions and meanings of what a public space is from different stakeholders, which, in practice, can allow us to prioritise the arrangements needed to achieve quality in the public space [32]. The aspiration of public spaces can enhance the empowerment of residents and other stakeholders when improvements based on their needs are achieved,

and these vary among cultural, economic and social contexts [3]. Lastly, the conditions and functions of public spaces are relevant to understanding what physical conditions they have, what functions they offer and how they are used by users, being affected by the context where they are located [33]. Following the context, five dimensions are analysed to identify how the management is being perceived by different stakeholder groups to achieve quality in the public space. In this research, we drew on an adapted version by [31] and incorporated social participation as a key dimension to be encompassed, aiming to understand the effectiveness of involving local residents in PDMPS in the Latin American context (see Figure 1). Doing so highlighted the need to understand the involvement and transfer of responsibilities to social actors and how they communicate [4], which demonstrates its suitability for the research at hand.

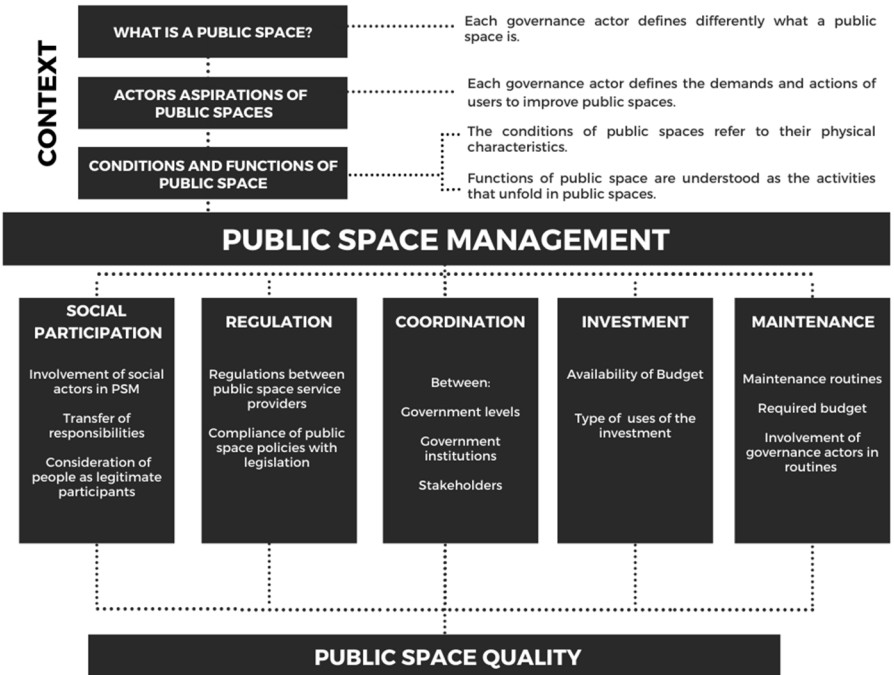

**Figure 1.** PSM dimension for the Latin American context. Source: [5].

3.1.2. Democracy Diagram

The democracy diagram is a conceptual framework based on an adaptation of the democracy cube, which aims to address social participation in public space management. The original democracy cube analyses institutional choices according to participatory mechanisms, with communication, authority and power, and participants as its three axes. The model seeks to understand how participatory projects in the planning process are developed [34]. Here, we use the adapted version of the democracy cube—the democracy diagram, which was adapted by [12]. The democracy diagram aims to understand how PDMPS decision-making processes consider resident inputs, analysing the number of actors involved, the level of communication and the level of decision making achieved. The authors of [12] also recommend utilising innovative solutions (i.e., ICTs) to support participatory processes, which justifies their use in this research.

Informed by the diagram, in this research, the communication achieved through ICTs is defined as the understanding of the intensity of interactions between one or more actors in a discussion of a particular issue [9,10]. The levels of communication are determined by a growing intensity scale: no communication, listen as a spectator, express preferences, develop preferences (co-creation), deliberate and negotiate, and deploy technical expertise. By offering different communication channels and possibilities for different degrees of communication, ICTs can boost participatory processes and public involvement and, thus, enhance project legitimacy and acceptance [11]. Just as ICTs have been used to improve

communication processes, they have also created avenues for empowering residents in decision making. The authority and power dimensions explore how participant inputs are considered or linked to the decision-making process or actual outcomes in real-life projects [10,12].

### 3.1.3. Research Steps

In the initial step of this research, we analysed studies that used ICTs to support PDMPS. We obtained fifteen research papers for further analysis using a systematised literature review. Section 3.2 covers the focused literature review in detail.

In the second step, we conducted 21 semi-structured interviews. The objective of the interviews was to acquire insights into the governance actors' perspectives on how ICTs are currently used to support participatory processes, the challenges encountered, and whether these challenges differ from those identified in the literature (see Table A1 in the Appendix A). Nineteen semi-structured interviews were conducted in our selected case study areas between November 2019 and January 2020, and two more were carried out in August 2021. The interviews were held, transcribed and coded in Spanish. Parts of the interviews were translated into English to provide illustrative quotes. The interviewer recorded 20 interviews (24 h of audio recordings) during fieldwork and took extensive notes during one interview, as the interviewee did not wish to be recorded (see Table A2 in the Appendix A).

The data from the transcripts were organised in a spreadsheet according to the four groups of actors, which we discuss in more detail in our Findings section (Section 4). Next, a table of twelve aspects was elaborated to analyse the use of ICTs that support social participation, which is further explained in Section 4.2. In the second stage, the information collected from each group of stakeholders was organised based on a table of aspects of ICTs that support social participation in PDMPS, this is explained in Section 4.3.

### 3.2. Literature Review of Studies on ICT Use to Support PDMPS

To create an overview of current research about the use of ICTs in participatory processes, the focused literature review targeted studies that investigated PDMPS processes. An additional purpose of the review was to obtain information for the analysis of the semi-structured interviews.

We searched journals using two different databases (Web of Science and Scopus), with a combination of the three key concepts as search terms: 'public space', 'social participation' and 'information and communication technologies'. The exploration of alternative databases returned similar research, including papers that were not selected because of their different foci. We also used hyponyms such as 'public park' or 'public participation' in the search. Considering the substantial improvement in digital accessibility and the use of mobile devices (e.g., smartphones and tablets) from 2012 onwards, the search focused on papers published in the last ten years [35,36]. Table 1 provides an overview of the key concepts used in the two aforementioned databases, including hyponyms and acronyms of the same key concepts. The exact queries can be consulted in Table A3 in the Appendix A.

**Table 1.** Key concepts, hyponyms and acronyms used for the literature review.

| Key Concepts | Hyponyms and Acronyms |
|---|---|
| Public space | Green infrastructure, green space, public park |
| Social participation | Public participation, participatory mapping, citizen science |
| ICTs | PPGIS |

The next step involved an exhaustive review of 115 abstracts, where we inductively identified papers relevant to our research, leaving aside studies outside our focus related to PDMPS (i.e., studies on telecommunications, political participation, the evaluation of participatory processes and smart cities were excluded). From the pool of 115 research papers, 15 papers were selected as closely related to the scope of this research (see Table A1).



We followed the work of Snyder (2019) on the literature review as a method, looking for the contribution of papers with evidence that can inform policy or practice [37]. We aimed to find articles that are related to our research scope yet broad enough to provide adequate information regarding the use of ICTs to support the participatory process for PDMPS (see Table A1 in the Appendix A). The following information was obtained from each article: (i) the concept or approach used; (ii) the involved stakeholder group(s); (iii) the purpose of use; (iv) which of the three phases of planning, design and maintenance from the PSM framework was used; and, in relation to the democracy diagram, (v) the level of communication supported and (vi) the decision making supported. Additionally, we also explored (vii) the hardware used, (viii) the software used, (ix) the connectivity requirements and (x) the highlighted challenges.

This section focuses primarily on the academic literature documenting the use and application of ICTs in participatory processes that have been carefully identified for analysis. Nonetheless, using ICTs in a non-academic context may be more diversified and multifaceted in day-to-day practice. An example in the Mexican context is the evaluation of air quality by the Ministry of Environment of Mexico City, which proposes diverse communication channels with residents through social media, such as Facebook and Twitter, to analyse sources of contamination that affect air quality in public spaces [38]. In addition, through the update of the new urban development programme (2024–2028), the Municipal Planning Institute of Puebla is using social media to share digital surveys among residents using platforms such as Google Forms [39].

### 3.3. Interviews: Actors and Analysis

We conducted semi-structured interviews with stakeholders who have a say in policy and decision-making processes in PDMPS in Mexico City and in Puebla. We also looked for actors with basic knowledge of the technologies and their application in daily work. The respondents were chosen based on previously conducted public space research [4,40,41] and divided into four types:

- Government: This group incorporates institutions and organisations that deal with public places and promote participatory processes at the federal, state and local levels [40,42].
- Non-governmental organisations (NGOs): This group is typically established by local residents with similar viewpoints to address a particular urban issue [43,44]. For this paper, we only considered NGOs that work on public space issues and serve as advocates for users and local residents.
- Architecture/urban planning consultancy firms: These companies draft project proposals in response to government requests, such as tender invitations, and implement government projects. Governments frequently recruit consultancy firms because they lack the required technical or organisational capacity. Their contracts occasionally call for the establishment of participatory procedures [45].
- Academia: Local universities frequently offer scientific insights on the conditions of local areas and are invited to consult as subject-matter experts in decision-making processes [41,46]. They frequently interact with local residents at large through academic research, outreach and media contributions and often are requested to participate in participatory processes [47,48].

### 3.4. Selection of Case Study Areas

We focused on Mexico City and Puebla, both located in the megalopolis of central Mexico and within close geographic proximity. Both cities face similar PDMPS challenges, namely, the lack of security and attention from local administrations to improve the conditions of public spaces [4,29,49,50].

These cities were selected because their agendas at the national and municipal levels attempted to foster social engagement in PDMPS-related issues through government institutions, with particular emphasis on the implementation of digital technologies to

store, process and distribute information [51,52]. Mexico City created institutions that focused on PDMPS and explored using ICTs for participatory processes. For example, the Ministry of Public Space of Mexico City (2008–2019) and the Laboratory of Mexico City (2013–2018) developed strategies to interact and engage with local residents through web-based platforms, social media and physically interactive workshops [53,54]. The Municipal Planning Institute of Puebla tried to create a web platform to present spatial information to the general public, including a catalogue of public spaces [55]. These are sparse and innovative efforts, as the current guidelines created by government institutions still promote analogue participatory methods [56].

## 4. Findings

### 4.1. ICT Support of Social Participation: Literature Review

The literature review primarily uncovered ICTs based on using GIS for data collection through surveys with mobile or web applications, together with social media platforms and 2D and 3D modelling tools. To understand how ICTs have supported social participation, we summarised the findings of the fifteen highlighted papers (see Table A1 in the Appendix A). The analysis is presented through the PSM framework and the democracy diagram for PDMPS [4,12].

For the planning phase, ICTs allow participants to comment and provide their opinions for context analysis and decision making in planning public spaces (see Table A1 in the Appendix A), primarily through field or online data collection and visualisation. They enable participants to see other actors' perspectives on public spaces and documents and share the current conditions in the urban area [5,57–62]. Most of the ICTs reviewed are used by government actors to communicate with local residents to allow them to express their preferences. The decision-making process is mostly in the hands of government actors, who later inform the residents about the results [61,63–65].

For the design phase, we found ICTs for collaborative ideation, design and visual representation. Ideation is the process of generating ideas that could further be developed through conceptual designs and visually represented via quick prototypes using 2D/3D models or other visualisation tools [57,66,67]. There is a trend of applying VR or gamification methods, a process where game design mechanics are used in urban planning to engage with users in a non-game context [68]. Gamification has been used to collaboratively design urban spaces between local residents and government practitioners to visualise, discuss and make decisions about future projects with decision-makers [17,69,70].

For the maintenance phase, the reviewed literature recommended using ICTs that allow the long-term support and maintenance of existing public spaces. The ICTs need to enable stakeholder input to monitor, evaluate and update the conditions of public spaces, allowing room for discussion and directing efforts to where they are most needed [5,21,60]. The level of decision making achieved by local residents in the maintenance phase mainly reaches public consultations. ICT communication channels are deployed to enable residents to express their perception or preferred use of public spaces [21,62,64].

It is often mentioned that the achieved outcomes do not yet match the anticipated benefits of articulating the needs and aspirations of the local residents. This is mostly due to a lack of integration, inconsistency in the verification of data collected, a lack of political will and investment, and a lack of social visibility of the use of ICTs to support participatory processes [21,58,62,64,67,71].

### 4.2. ICT Aspect Matrix

The literature review revealed seven useful aspects to understand how ICTs are used to support participatory processes (see Table A1 in the Appendix A). We added more aspects informed by the research of Bratunskins et al. (2020), who presented a classification of aspects when considering digital tools for urban regeneration [72], and the work of Hanzl (2007), who analysed the use of ICTs as an experimental process for urban planning [73]. Both studies analysed the purpose of the use of ICTs in participatory

processes. In order to allow a more detailed analysis of the tools used in practice for PDMPS, we include (1) the type of visualisation supported; (2) access based on use and modification restrictions; (3) technical capacity; (4) requirements for spatial literacy; and (5) the participatory interaction setup. In total, twelve aspects are applied in the case study (see Table 2).

**Table 2.** ICT (hardware, software and purpose) support of social participation in PDMPS—research design.

| Aspects | | Details of Inquiry |
|---|---|---|
| Purpose of use | | What was the purpose of the use of the digital tools used? (collaborative mapping, creating a survey, sharing information, allowing digital drawing, downloading data, etc.) [72,73]. |
| PSM framework | | In which of the phases of planning, design or maintenance was a specific ICT used to support participation? |
| Democracy diagram | Communication level supported | The achieved level of communication for which the ICTs were used (e.g., to express a preference or collaboratively develop a preference) [12,34]. |
| | Decision-making level supported | The decision-making level for which the ICTs were used, related to the dimensions of the authority and power of the democracy diagram [12,34]. |
| Tools used | Hardware | Whether portable or desktop devices were used (i.e., laptop, smartphone or tablet) to support discussion and communication among participants and whether they were combined using an interactive input device, such as a maptable that requires a desktop device, or a portable VR headset, such as Oculus, that does not need a connection to a desktop device [74]. |
| | Software | Whether a specific digital tool (software, applications, web platforms) was used [72,73]. |
| | Types of visualisation supported | Whether digital visualisations were used to engage with stakeholders to communicate information about spaces, such as maps developed with GIS software, photographs, video, rendering images from a 2D or 3D architectural model, 3D simulation of virtual environments or interactive web maps [75]. |
| | Use and modification restrictions | Whether the digital tool has licensing restrictions or fees [76]. |
| | Technical capacity | Referring the extent to which a digital tool is easily available and accessible to the public sector [19,77]. |
| | Requirements for spatial literacy | Referring to the level of knowledge to use digital tools based on their properties to communicate, discuss and provide solutions to urban issues [77,78].<br>■ Low—basic skills required to distinguish spatial elements;<br>■ Medium—adept at relating and transforming spatial data on urban issues;<br>■ High—educated in solving complex situations with spatial reasoning using digital tools. |
| | Connectivity requirements | Whether the ICT requires constant or frequent online connectivity:<br>■ None—the complete participation session can be performed offline;<br>■ Low—only small data volumes, covered by smartphone data bundles;<br>■ Medium— stable broadband access with low latency;<br>■ High—High-speed broadband connectivity. |
| | Participatory interaction setup | Whether ICTs require a shared physical location for participation or a virtual environment such as a web-based platform, and whether the interaction has to be synchronous (real-time) or asynchronous (over a period of time) [74]. |

### 4.3. ICTs Used for PDMPS in the Case Studies

The findings are presented according to how the interviewed actors (government, NGOs, academics and architecture/urban planning firms) discuss (1) the use of ICTs to facilitate social participation in PDMPS; (2) to what extent the various actors involve local residents based on the proposed aspects of ICTs for PDMPS; and (3) the challenges and opportunities they faced when trying to introduce ICT-enhanced forms of social participation.

Table 3 provides an illustrative example of the evidence obtained from one of the four groups of actors and how we organised the data in our research design. The data obtained came from the semi-structured interviews conducted during our fieldwork. The same data processing was followed for the other three groups of actors.

**Table 3.** ICT (hardware, software and purpose) support of social participation in PDMPS—government officials.

| Level of Communication and Use of ICTs: Government Officials | | | |
|---|---|---|---|
| **PSM Framework** | **Planning** | **Design** | **Maintenance** |
| Purpose of use | Spatial analysis and visualisation. | Creation of conceptual design and visualisation of a public space design to explore design alternatives. | Manage the location and characteristics of current public spaces in Mexico City and Puebla. |
| Communication level supported | Communication is among government institutions; with other stakeholders, they only inform about a project. | Internally and at a government level.No communication with social stakeholders, just with the private sector, as designs are outsourced to private companies. | No communication with stakeholders; they inform residents about the programme for public spaces. |
| Decision-making level supported | Internally without consulting other stakeholders. Local residents are consulted about their preferences, but this is not a common practice. | Decisions are made through revisions with architecture/urban consultancy companies. | Internally, without consulting other stakeholders. |
| Hardware | Desktop computer, tablet and smartphone. | Desktop computer. | Desktop computer. |
| Software | ArcGIS, QGIS, Google Earth and Mapillary | AutoCAD and SketchUp. | ArcGIS and Google Earth. |
| Type of visualisation supported | Maps and statistics. | 3D rendering. | Maps and a webpage. |
| Use and modification restrictions | Proprietary and open-source tools are used. | Proprietary. | Proprietary and open-source tools are used. |
| Technical capacity | In the three phases, relatively easy access. | | |
| Requirements for spatial literacy | High | High | Medium |
| Connectivity requirements | Medium | Medium | Low |
| Participatory interaction setup | In the three phases, participatory practices are carried out in a physical setting and through asynchronous collaboration among participants. | | |

*4.4. Mexican Context*

In the Mexican context, laws, policies and regulations aim to create conditions for more inclusive participatory processes and encourage the use of ICTs to support them [5,14,15]. The digitisation of government information is a priority, as some existing data sets are still in analogue formats (e.g., handmade maps, sketches and documents), and even new information is still collected in analogue forms. According to federal and local governments, these analogue processes need to be transformed into processes supported by digital technologies. However, government practitioners mentioned that there are still challenges in using ICTs in daily practice [16,17]. The discussion below outlines the interview findings for each stakeholder group and is structured by the main planning, design and maintenance phases.

4.4.1. Government Officials

For participatory processes, government officials mentioned using proprietary and open-source GISs (e.g., QGIS or ArcGIS) in all three phases of PDMPS. All stages require a

stable internet connection, which is usually available in government buildings. However, the connection speed is often limited to basic tasks, such as sending emails or looking for information in a web browser. It was also mentioned that all the tools require a professional/technical capacity, which can create a digital divide with local residents when using these tools independently. Some difficulties in their use could be due to factors such as age, physical limitations or poor ITC literacy. Furthermore, local residents are rarely invited to participate in the decision-making process.

At the federal level, government officials only participate as observers in community meetings and discussions with local residents, collecting feedback on notebooks, tablets or smartphones. The federal government responders mentioned that some participatory processes are currently outsourced to universities or private consultancy companies. Only social media was mentioned as used to present government projects and receive feedback from comments on the official government Facebook and Twitter accounts:

> " . . . *just the basic social media is used, such as Facebook, Twitter and WhatsApp, to promote a campaign or a meeting, but I do not know any technology tool that can solve a participatory process more effectively than how it is currently done on a daily basis face to face.*" (Interview with a government official at the federal level.)

For the planning phase, the local governments in the cities of Puebla and Mexico City mentioned using social media through a desktop computer. In addition, mobile-based applications, such as Google Earth or Mapillary, allow them to use street-level images and mark locations or map elements of the city, such as wastebaskets, benches or bicycle parking. Their primary use is for spatial analysis and the creation of development plans, including existing and possible new public spaces. The information collected is used to create materials that can be printed and used in a workshop or a focus group with local communities. While all tools require only low- to mid-cost hardware, they are a mix of both proprietary and open-source. The typical interaction setup occurs within a physical location, with asynchronous collaboration among stakeholders when a digital tool is employed. Usually, feedback is not collected digitally.

For the design phase, local government respondents in both cities mentioned using architectural drawing software, such as AutoCAD or SketchUp, for 3D modelling to present projects and proposals for public spaces on official websites and during focus groups. All the tools are proprietary. The direction of communication with other stakeholders is mainly between the local government, the architecture/urban planning consultancy firms that design the public space project, and other government institutions. Local residents and NGOs are usually not considered in the design phase but are only consulted for their opinions on previously designed projects. In addition, while the government makes the final decisions, it does not create public space projects but outsources them to architecture/urban planning companies. The devices used are desktop computers. Software tools are used to analyse the projects currently being supervised within government institutions. In some cases, 3D-rendered images of the projects are shared on official government communication sources (websites and social media). The 3D-rendered images and blueprints are also printed to present a project in focus groups with local residents or other stakeholders, such as academics.

For the maintenance phase, local government respondents from Puebla mentioned using a GIS and I-TREE ECO and ARBOTOM software to collect and manage information about the conditions of trees. The Municipal Planning Institute of Puebla presented a web-based GIS system called SIGEM (SIGEM is the acronym of 'Sistema de Información Geográfico Municipal', which, translated into English, means the municipal geographic information system) to visualise the availability of public spaces in an effort to increase transparency. The information is publicly available on the official website, which also collects comments and suggestions from local residents. However, the contribution to decision making reaches only the consultation level. It is the only official source of information publicly available (see Figure 2). In Mexico City, no specific tool was mentioned, only a

web platform presenting basic information about a federal programme on improving the urban conditions of several areas in the country.

**Figure 2.** Availability of public spaces, illustrated using SIGEM (2018). Source: [79].

4.4.2. Non-Governmental Organisations

The NGOs are social organisations that have already conducted co-creation processes with local residents. Similar to the local government interviewees, the NGOs mentioned using proprietary and open-source GIS software (e.g., ArcGIS or QGIS) in all three phases of PDMPS. A stable internet connection is required to communicate the generated data with other stakeholders via applications, such as WhatsApp or Facebook, except in areas with no internet access. Desktop computers and mobile devices were utilised throughout all phases.

NGOs in Mexico City and Puebla City frequently use social media and web-based surveys during the planning phase to diagnose the current conditions of public spaces. The participatory processes mostly take place in physical locations in parks and streets or in indoor locations, both synchronously and asynchronously, with NGOs facilitating interactions. Residents usually express their needs without actually being involved in decision making. The software tools are mostly open-source and require an advanced technical capacity. Digital visualisations include maps, images and blueprints created with an open-source GIS or proprietary CAD software, which are later shared digitally or on paper with residents and other participants during workshops (see Figure 3).

In addition to GIS and 3D modelling software, they use Photoshop or Illustrator during the design phase to develop posters or illustrations and visualise public space ideas and maps of possible interventions. The level of communication is to express preferences on what residents and other stakeholders want to see in public space proposals using digital 3D-rendered images, maps and even 3D model prints. An NGO in Puebla uses 3D printers to create models to include visually impaired people in participatory processes. While they also use proprietary tools, NGOs in Mexico City favour open-source tools, despite the fact that they require a higher professional technical capacity. NGOs interact with stakeholders in physical locations through facilitated participation and sometimes present their models through social media and websites.

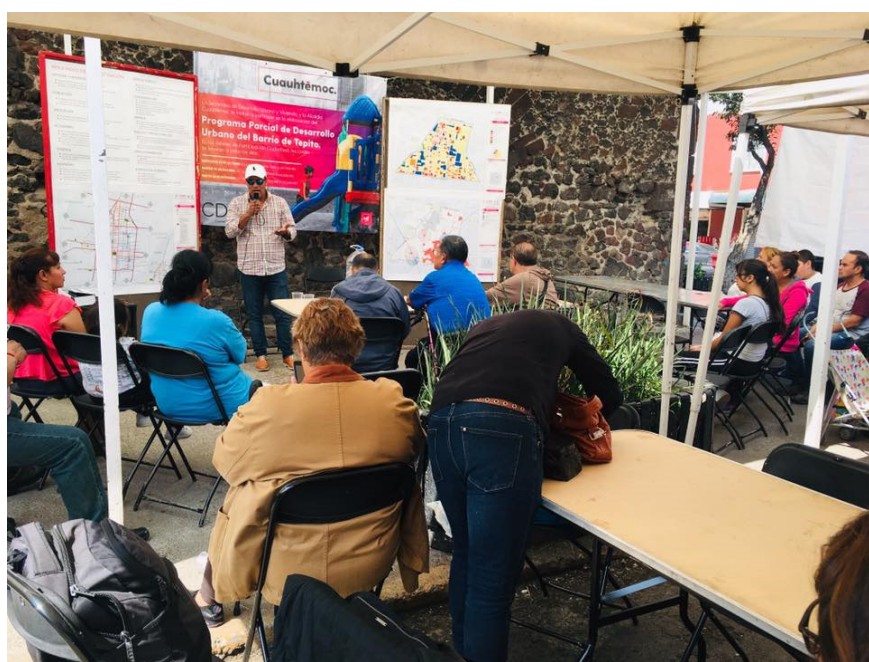

**Figure 3.** Participatory workshop to develop the local development plan for the Tepito neighbourhood (2018). Source: [80].

In the maintenance phase, organisations from both cities rely on web-based surveys to collect spatial information, create statistics on the current situation and prototype ideas for improving and managing public spaces. Communication efforts create a space for residents to express their needs and preferences regarding improvements in the physical conditions of public spaces. Usually, these exchanges happen through digital communication platforms such as WhatsApp. The organisations use the information collected in public consultations in their advocacy work to influence decision making.

### 4.4.3. Academics

When participatory processes are involved, the interviewed academics mentioned using GIS software in all stages. An internet connection is required to disseminate participatory activities among different stakeholders; however, academics have inconsistent access because of the limited broadband connections of Mexican universities. They rely on desktop computers and proprietary tools, which can be easily licensed through the university, during all stages.

In the planning phase, GIS is used to support the development of urban plans and diagnose the conditions of public spaces. The level of communication is that residents sometimes express preferences and listen as spectators in neighbourhood meetings, focus groups or walking interviews. No decisions are made, and the aim is to inform local residents about current urban projects. To organise participatory processes, academics also communicate through social media with other stakeholders, such as local NGOs. They disseminate posters and banners for future activities via social media networks, especially Facebook (see Figure 4). Regarding public space issues, academics mentioned using printed maps and surveys or web-based surveys, usually in physical locations and through synchronous collaborations where academics or students facilitate a participatory process as part of university projects or research.

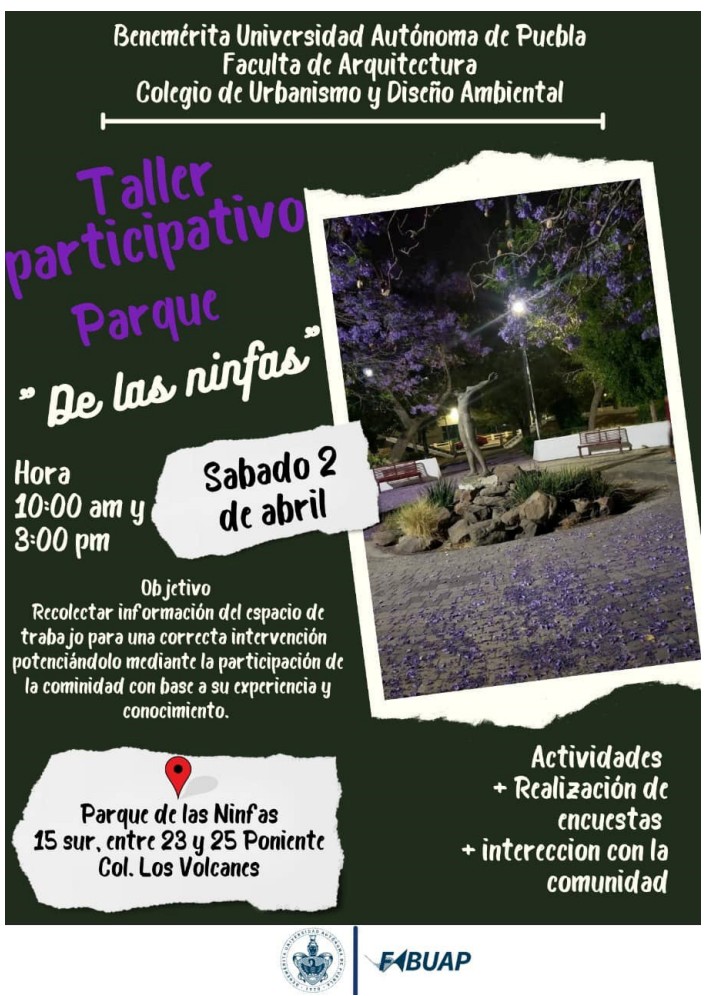

**Figure 4.** Facebook invitation poster for a participatory workshop by academics and an NGO in Puebla City. Source: [81].

Academics use GIS (e.g., ArcGIS and QGIS) and 3D modelling software (e.g., SketchUp and AutoCAD) to visualise public space proposals developed by bachelor's or master's students. Some tools are used to generate awareness and collect LSK among stakeholders in the participatory process and to represent how they imagine a new or revitalised space, but no decisions are made. The level of communication mentioned is that academics actively engage in discussions about one or more projects during neighbourhood meetings or focus groups, collecting insights and perspectives through note taking. However, most of the participants may simply observe these discussions as spectators. Academics present the public space proposals using digital 3D images, mainly in a physical location, and facilitate participatory processes but do not co-design with the residents. Advanced technical skills are required to use these tools.

Academics are less involved during the maintenance phase, and just one participant mentioned using GIS to measure the number of pedestrians walking in public spaces and the development of a catalogue of heritage using a desktop computer. There was no communication with other stakeholders, and no decision-making processes were mentioned.

4.4.4. Architecture/Urban Planning Consultancy Firms

Architecture/urban planning consultancy firms rarely take part in participatory processes. In all phases, proprietary and open-source software and broadband internet access are required. Desktop computers, smartphones, tablets and even small unmanned aerial vehicles (UAVs) are engaged, and most of the tools require complex technical skills.

For the planning phase, they have access to more complex ICTs requiring paid licences, for example, web-based and mobile survey services (e.g., Google Forms or SurveyMonkey), video communication tools such as Zoom and storage cloud services such as Dropbox or Google Drive. This group of actors uses GIS software to conduct spatial analysis, organise data collection and develop diagnostics for urban areas. There was no mention of implementing participatory approaches for gathering input from local residents and establishing effective communication channels. Instead, communication was primarily conducted with key decision-makers, such as clients, real estate firms and government agencies, who commissioned public space projects. The quote below illustrates to what extent the knowledge held by local residents is actually recognised by professional experts such as architects. The inclusion of local residents in PDMPS requires an appreciation of the validity of each other's knowledge, particularly in cities with emerging economies, as stated by [13].

> *"Using technology is limited for some people; not everyone knows how to use it. This is due to the generational gap, the costs of having a device as a smartphone or the lack of knowledge how to use it . . . participatory results can be unpredictable."* (Interview with an architecture/urban planning consultancy firm.)

For the design phase, they use drawing and 3D modelling software (AutoCAD, SketchUp, Revit or Lumion) and graphic design software tools (Photoshop or InDesign). These tools are used to present public space proposals to their clients, usually government institutions or the private sector, such as real estate companies. No communication was pursued with other stakeholders, and there was no mention of any participatory setup. Digital 3D images such as renderings or 3D virtual tours were used to present the public space proposal. All tools are proprietary and sometimes need external support, for example, photorealistic rendering.

For maintenance, they only mentioned using GIS software and devices to identify vulnerabilities of uses in public spaces, create an inventory of vegetation in public spaces and analyse possible interventions in existing public spaces. There was no communication with other stakeholders in the maintenance phase, meaning that a participatory setup was not necessary, and no decision-making processes took place.

### 4.5. Using ICTs for PDMPS: Challenges and Opportunities

All actors mentioned ICTs as an opportunity for PDMPS, highlighting four main reasons:

1.  ICTs can improve data collection in the field and online, analyse and process information about the current conditions of public spaces more quickly and more efficiently and share that information with other relevant stakeholders.
2.  ICTs open new opportunities for social participation and involvement by using other digital communication channels, such as social media, web-based surveys or other digital platforms that allow decision-makers to receive feedback from other stakeholders and share relevant information.
3.  ICTs help stakeholders to inform, raise awareness and educate through visual information (e.g., maps, renders, 3D videos, infographics and websites).
4.  ICTs can lower administrative costs with open-source or free-to-use digital tools. While this option reduces costs by eliminating proprietary software licences and membership fees, it requires higher technical skills and knowledge.

Nevertheless, ICTs also pose various challenges. For example, government officials still hesitate to support participatory processes with ICTs because of the required technical capacity and access to devices (e.g., smartphones). One interviewee from the federal government mentioned that using ICTs without a proper social vision can benefit private or commercial actors, especially real estate companies, at the cost of social improvement. The available public space data are used by private companies for commercial reasons rather than to improve the conditions of public spaces for local residents. Another interviewee from the local government of Puebla City mentioned that the benefits of ICTs are not

yet clear. One NGO said that they do not use ICTs in participatory processes, as they have yet to explore its use in the field. Due to cybersecurity concerns, they prefer to use analogue methods (i.e., printed images and blueprints). We found that academics are less accustomed to using ICTs for PDMPS and still prefer to use analogue methods. Academics rely on students when conducting research that demands greater technical capacity. The architecture/urban planning consultancy firms felt that technology use will gradually increase and that we will reach a point where no one will notice the change. Nevertheless, accessibility will be a problem due to cost barriers:

> *"There is still a focus on commercial technologies, which implies costs that are not cheap . . . . There is a need to build conditions to develop projects using more open-source technologies."* (Interview with an academic in Mexico City.)

**5. Discussion**

This paper focuses on understanding how ICTs can support social participation in PDMPS. During the data analysis, the table of aspects (see Table 2) helped inductively describe the phase of planning, design and maintenance achieved through its use, as well as the level of communication and power. The discussion below is structured by the main phases of planning, design and maintenance.

*5.1. Planning*

Many developing countries across the globe use ICTs during PDMPS to collect LSK. According to our literature review, PPGIS usually involves web-based platforms or mobile apps to bridge the gap between planners and residents [5,82]. However, in our case studies, most of the actors use desktop GIS software asynchronously for data analysis during the planning phase. Although all interviewed stakeholders used web-based surveys for data collection, none of the tools adopted collect data with spatial attributes or location. Academics commonly mentioned analogue techniques (pen and paper). Despite important efforts to use ICTs during the planning phase, the actual outcomes have fallen short of the potential benefits outlined in our literature review, such as increasing participation, providing visual aids, increasing transparency or collecting insights regarding the needs and desires of local residents [61,65]. The ICTs in our case study were primarily used to present information via unidirectional communication. GIS or 3D modelling software was mainly used to produce maps and images that were later shared to inform residents about public space projects. In addition, government institutions use social media as a one-way communication channel. However, all interviewed actors mentioned the lack of synchronous participatory approaches with residents that involved bidirectional information flows.

Researchers increasingly utilise gamification methods as a means to enhance participation and legitimise decisions related to urban planning, also highlighted in the literature review [66,83]. We found that isolated gamification experiments utilising proprietary mobile apps, such as Pokémon GO, have been conducted in Mexico City by the federal government to explore public spaces with the participation of local stakeholders [54]. Local governments in other contexts, such as Japan, have already implemented exercises incorporating gamification models to solve urban issues [84]. As Mohammed and Hirai (2021) discuss, research is critical to noticing the global context potentials and challenges of gamification in practice for enhancing public spaces.

> *" . . . applications such as Pokémon GO is an example of technology used in the public space, you can look for public spaces where people gather to collect the Pokémon's."* (Interview with a government official at the federal level.)

*5.2. Design*

Interviewees commonly mentioned using images to convey ideas to the public in the design phase, namely, printed maps, blueprints or 3D modelling images. Some researchers mentioned that visual communication in urban design and planning is important to communicate current or future urban spaces between creators (i.e., urban planners)

and recipients (i.e., social organisations) [57,83]. Drawings, maps, photographs and 2D or 3D city models are among the digital visual representations common in participatory approaches [66,85].

Despite the potential benefits of ICTs to improve communication, our case study indicates that ICTs do not yet meet the conditions necessary for government practitioners, academics and NGOs to innovate in the design phase. The challenges of engaging with local residents, inadequate infrastructure (such as the absence of computers with 3D visualisation capabilities) and a shortage of technical expertise feed the digital divide, as also reported in the literature [18,19,83].

Architecture/urban planning consultancy firms invest more time and money in high-quality images, exploring new software for photorealistic renderings and 3D virtual tours. However, visual aids are mainly used in presentations with their clients, not in participatory processes. Other ICTs mentioned in the design phase include 3D printers, collaborative open-source apps or small UAVs to communicate and collaborate with other stakeholders. These tools are widespread in the Do-It-Yourself (DIY) movement, which works on increasing ITC literacy and technical skills via indoor spaces called Fablabs, hackerspaces or repair cafés. These spaces aim to solve societal problems using technology, including urban issues [86]. There have been attempts to create these spaces in Mexico City and Puebla through government institutions or NGOs. However, a change of administration in local and federal governments breaks the continuity of these initiatives, usually due to political differences or changes in government strategies [4,87].

Current research explores the use of interactive or immersive digital tools, such as 3D modelling, augmented reality or a web-based GIS, using hardware such as desktop computers or mobile devices (smartphones, tablets or VR headsets) [57,66,67]. However, we found that, in Mexico, spatial tools are used through desktop computers in the design phase but are not used to collaborate or communicate design choices in participatory processes. There are opportunities to strengthen the use of ICTs in the design phase to seek residents' approval, as research aims to include local residents in decision making, and interactive tools enable improved communication and interaction [5,15].

*5.3. Maintenance*

The literature review shows the frequent use of ICTs during the maintenance phase. However, some challenges remain, such as the lack of investment, the need for innovative policymaking to promote the use of ICT, data inconsistency or the absence of data that could provide insights into how to maintain public spaces in optimal conditions [21,58,64]. These shortcomings also emerged in the case study. All interviewees noted the lack of participatory processes that engage residents. Notably, government stakeholders inform residents about maintenance works but do not invite them to co-decide on maintenance-related issues.

As demonstrated by the literature review, ICTs have addressed some challenges by supporting participatory processes as mediating technologies [5,6]. In our case study, the actors shared that, despite attempts to incorporate ICTs to support participatory processes in PDMPS, the potential of these technologies remains largely untapped. They ascribe this underutilisation to the lack of advanced technical capacity among government practitioners and insufficient experience with the right tools to support participatory processes and consider resident feedback. These challenges were also mentioned in the literature (see Table A1 in the Appendix A).

**6. Conclusions**

This paper enhances the understanding of how ICTs can support social participation in the planning, design and maintenance of public spaces with three important insights. First, the use of ICTs is increasingly evident in PDMPS, yet its implementation to support participatory processes has yet to be fully embraced by the various stakeholders involved in decision making, particularly policymakers. Second, this paper reveals an installed base of

digital tools and devices used and the technical capacities available. However, despite these resources, applying ICTs to support participatory processes is still challenging because of the digital divide (especially among local residents), the lack of technical capacity in government agencies and the lack of data regarding PDMPS. Finally, social media such as Facebook or Twitter are revealed to be the only communication channels between local residents and the government, while other approaches, such as PPGIS or collaborative design for PDMPS, are still not common practice. This is a missed opportunity for the level of communication and of the authority and power achieved in the interaction with other stakeholders to generate dialogue and shared proposals for PDMPS.

In terms of methodological contributions, our research developed an ICT aspect matrix (see Table 2). The matrix consists of twelve aspects to be used as a methodological reference to identify and analyse the role of ICTs in supporting social participation in other geographic contexts with similar conditions, arrangements and challenges to Mexico [88,89]. Through its application in the Mexican context, we could evaluate the existing potential for the enhanced utilisation of ICTs at local levels, such as the benefits of communication between decision-makers and other stakeholders, especially between governments and residents. We argue that the matrix developed in this paper can also be of use for other contexts and governance levels. We have made this contribution more explicit in our conclusion section.

Given the recent exploration of using ICTs to support participatory processes in urban planning issues [58,63,83], our research contributes to this scientific body of knowledge, with an added focus on the planning, design and maintenance of public spaces [5,8,62,66]. We argue that the visibility and awareness of ICTs to support participatory processes in PDMPS need to be increased among the different actors who have a say in decision making. To ensure better-informed decision making, permanent communication channels should be maintained with residents, documenting their needs and aspirations. Although all actors interviewed for our case study see the use of ICTs as an opportunity, the challenges presented at the beginning of this section continue to inhibit their application in daily practice. The lack of updated data is one of the biggest problems in decision-making processes and social participation.

In the international context, our findings contribute to the recent exploration of innovative approaches that could enhance social participation in PDMPS but also to the uncertainties in their use in practice. Moreover, as seen in the Australian case, there is a need to find innovative methods for developing co-creative processes and engagement strategies that include multiple stakeholders while attempting to avoid false participation expectations through digital platforms [90].

Nevertheless, as this research shows, these tools are rarely implemented in practice in all phases of PDMPS, as the government still has not created the conditions for inclusive participation. We observed that even if ICTs are used, government institutions have exclusionary practices for data collected by local residents and key stakeholders, mostly due to the lack of opportunities and the provision of institutional spaces to discuss topics related to PDMPS. We also encountered challenges comparable to those reported in the European context, specifically that human and technical resources are still required and that the requirements for time invested are too high for planners and decision-makers to benefit from using ICTs [8,24]. Without the institutional embedding of participatory processes, using the 'newest' ICTs will be useless, and the voice of local residents will remain ineffective for decision-making processes.

The results underscore the potential for the enhanced utilisation of ICTs at both the local and international levels, such as the benefits of communication between decision-makers and other stakeholders, especially between governments and residents. Previous research stated that ICTs have recently been used to create new communication channels, enable stakeholder collaboration and broaden the influence of different groups in decision-making processes [5,60,82,91]; thus, we acknowledge the potential for the adoption and implementation of ICTs to enhance participatory processes in the everyday practice of PDMPS in Latin American and the international context. Yet, more research needs to

be conducted to explore their implementation to potentially be utilised in future urban practices for developing public spaces.

We recommend that government practitioners take a more proactive approach to advancing participatory processes in PDMPS in Mexico—throughout all phases. The available technologies provide a good basis to interact more actively with local residents and collect LSK.

This research's limitations include the potential lack of reliability concerning some of the interviewees' responses since some stakeholders reported a lack of expertise or knowledge in using ICTs. In addition, we were unable to corroborate the use of specific digital tools mentioned by government actors, as some of the projects in which these tools were supposedly utilised are not publicly accessible. In particular, projects originating from government agencies, namely, the Authority of Public Space or the Laboratory for the City, both from Mexico City, which were disbanded in 2018, and official public records of the mentioned projects could not be corroborated [92,93].

Further research could explore broader case study areas at the national level and expand the actors who have a say in participatory processes. Education in planning is an effective approach to motivate students to engage with local residents and receive feedback through ICTs. An example is the implementation of the visual presentation of proposals and evaluation by residents in participatory processes using virtual reality and 3D models in Tonalá, Mexico [6].

As this research was conducted during the COVID-19 pandemic, the shift to remote work may have changed the way participatory processes are generated in the case study areas, and specific niche tools could have become available.

**Author Contributions:** Writing—Original Draft Preparation, Conceptualisation, Methodology, Investigation, Visualisation and Funding Acquisition: Sergio Alvarado Vazquez; Supervision, Conceptualisation, and Writing—Review and Editing: Ana Mafalda Madureira; Supervision, Conceptualisation, and Writing—Review and Editing: Frank O. Ostermann; Supervision, Conceptualisation, Writing—Review and Editing, and Project Administration: Karin Pfeffer. All authors have read and agreed to the published version of the manuscript.

**Funding:** This research was funded by the Consejo Nacional de Ciencia y Tecnologia de Mexico (CONACYT) under grant no. 2018-000009-01EXTF-00207. The authors, therefore, acknowledge and thank CONACYT for technical and financial support.

**Institutional Review Board Statement:** The study was conducted in accordance with the Declaration of Helsinki, and approved by ITC Ethics Committee of the University of Twente. Reference:2020020301. Date of approval: 3 February 2020.

**Informed Consent Statement:** Informed consent was obtained from all subjects involved in the study.

**Data Availability Statement:** The data presented in this study are available on request from the corresponding author. The data are not publicly available due to privacy restrictions.

**Acknowledgments:** We want to thank all the people who so generously offered their time and expertise to carry out this research during fieldwork in Mexico.

**Conflicts of Interest:** The authors declare no conflict of interest, and the funder had no role in the design of the study; in the collection, analyses or interpretation of data; in the writing of the manuscript; or in the decision to publish the results.

## Appendix A

**Table A1.** Review of literature on ICT support of participatory processes.

| Authors | Concept/Approach | Stakeholders | Purpose of Use | PSM Framework | Communication Level Supported | Decision Making Supported | Hardware | Software | Connectivity Requirement | Challenges |
|---|---|---|---|---|---|---|---|---|---|---|
| [57] | Participatory urban planning | Local residents, government, private sector | Presenting a 3D model of an urban park to analyse to what extent a project meets the interest of local residents. | Planning and design | Express preferences | Collaborative planning and design | Gaming desktop computer | Unity | Not specified | Young people are rarely involved in planning processes. High-quality image rendering is still needed. |
| [58] | Management of urban public space | Local residents, government, private sector, academia | The use of ICTs to manage public spaces to achieve a smart city model. | Planning and maintenance | Express preferences | Inform | Smartphone application, CCTV surveillance cameras, wireless sensors | Smartphone applications (Smart Nation), internet | Internet connection | More investment and innovative policymaking are needed, and ICTs should be better utilised. |
| [66] | Gamification | Local residents, academics, professionals | Use VR in collaborative urban design to increase citizen participation. | Design | Express preferences | Collaborative planning and design | Smartphone, VR headsets | GAME4CITY | Internet connection | User interface is difficult to use. Participants get dizzy after using VR for a long time. Realism is still poor. |
| [59] | Public participation | Local residents | Use analytics of experiences of use of a park to measure its popularity. | Planning and maintenance | Listen as spectator | Inform | Any device with social media access (desktop computer, smartphone, tablet, etc.) | Social media application (Twitter), internet | Internet connection | Restricted to Twitter as a social media platform. |

**Table A1.** *Cont.*

| Authors | Concept/ Approach | Stakeholders | Purpose of Use | PSM Framework | Communication Level Supported | Decision Making Supported | Hardware | Software | Connectivity Requirement | Challenges |
|---|---|---|---|---|---|---|---|---|---|---|
| [60] | PPGIS | Local residents, government | Evaluate the perception of ICTs to create e-participation scenarios using PPGIS. | Planning and maintenance | Deliberate and negotiate | Decision making | Desktop computer | Internet platforms, digital surveys | Internet connection | Lack of knowledge on how to fill out the survey. Residents who do not know how to use the technology are excluded. |
| [61] | Participatory digital platforms | Local residents, government | Promote interaction between users and decision-makers on public space issues. | Planning and design | Express preferences | Inform | Bluetooth beacons, smartphone, tablet | Mobile application | Internet connection, Bluetooth | No widespread use of apps, and some data collected have errors or inconsistencies. |
| [21] | System design framework | Local residents, government | Analyse injustices associated with the revitalisation of public open spaces using an internet-based GIS. | Maintenance | Express preferences | Public consultation | Desktop computer | ArcGIS | Internet connection | Lack of data; integration with existing data is difficult, and encouraging people to participate is still a problem. |
| [64] | PPGIS | Local residents | A participatory classification of urban parks associated with park benefits. | Design and maintenance | Express preferences | Inform | Desktop computer | ArcGIS, SPSS | Internet connection | Data were inconsistent. |

**Table A1.** *Cont.*

| Authors | Concept/ Approach | Stakeholders | Purpose of Use | PSM Framework | Communication Level Supported | Decision Making Supported | Hardware | Software | Connectivity Requirement | Challenges |
|---------|-------------------|--------------|----------------|---------------|-------------------------------|---------------------------|----------|----------|--------------------------|------------|
| [83] | Gamification | Local residents, academics | Improve the lack of engagement of children and youth in urban planning. | Planning and design | Develop preferences in a co-creative setup | Collaborative planning and design | Desktop computer | Desktop computer, Minecraft | Not specified | Lack of basic infrastructure (e.g., computers and training) on how to use Minecraft. |
| [5] | PPGIS | Government, local residents | Participation through e-tools to engage residents in the planning and management of urban green infrastructure. | Planning, maintenance | Develop preferences in a co-creative setup | Collaborative planning and design | Desktop computer, smartphone | Web applications with GIS functionality | Internet connection | Lack of participatory processes, lack of inclusion, lack of social visibility. |
| [71] | Geolocated social media | Local residents | Analysing social media posts to understand the activity of use of public spaces. | Maintenance | Express preferences | Inform | Internet, desktop computer | Flicker and Twitter | Internet connection | Data still need to be verified. Supplementary research is needed. |
| [65] | Participatory mapping | Local residents government, academics | An application that reports problems and collects suggestions regarding public spaces. | Planning and design | Express preferences | Public consultation | Smartphones, desktop computer, internet | Web applications, Miramap | Internet connection | Interoperability with other tools should be expanded. Additional experimentation processes are needed. |

**Table A1.** *Cont.*

| Authors | Concept/ Approach | Stakeholders | Purpose of Use | PSM Framework | Communication Level Supported | Decision Making Supported | Hardware | Software | Connectivity Requirement | Challenges |
|---|---|---|---|---|---|---|---|---|---|---|
| [67] | Participatory planning | Local residents, government | Monitoring the use and activity of people in public spaces and documenting the physical settings of public spaces. | Planning and design | Express preferences | Public consultation | Sensors, desktop computer | 3D modelling tools, social media, web applications, remote sensing tools | Internet connection | Data need to be verified; smart city study needs to expand by measuring the quality of places in public spaces. |
| [62] | PPGIS | Local residents, government, professionals | A survey to identify the type and locations of urban park benefits. | Planning and maintenance | Express preferences | Decision making | Desktop computer | ArcGIS 10.2 | Not specified | Insufficient policies to support the use of PPGIS in cultural ecosystem services in public spaces and limited incidence in decision making. |
| [82] | PPGIS | Local residents, government | A web-based PPGIS to gather citizen data on visitor behaviour in Helsinki's Central Park. | Planning and maintenance | Express preferences | Decision making | Smartphone, desktop computer | MyDynamicForest, social media | Internet connection | More work is needed to address data heterogeneity and spatial accuracy and assess data quality. |

**Table A2.** Stakeholders interviewed during fieldwork in Mexico.

| City | Institutions | Type of Actors | Interview Date |
|---|---|---|---|
| Puebla City | Ministry of Mobility of Puebla | Local government | 13 November 2019 |
| | Municipal Planning Institute of Puebla | Local government | 14 November 2019 |
| | Mayor of the Romero Vargas District | Local government | 20 November 2019 |
| | Authority of the Historic Center of Puebla | Local government | 21 January 2020 |
| | Instituto de Ciencias Sociales Y Humaniades of the Autonomus University of Puebla (BUAP) | Academic | 11 November 2019 |
| | Faculty of Architecture of the Autonomous University of Puebla (BUAP) | Academic | 14 November 2019 |
| | College of Planners and Environmental Designers of the State of Puebla | NGO | 14 November 2019 |
| | Re-Genera Espacio | NGO | 15 November 2019 |
| | Entorno Paisaje | Private | 21 November 2019 |
| | Proyectos y Planeacion Integral S.A. de C.V. | Private | 21 August 2021 |
| | Servicios de Consultoria Urbano Ambiental | Private | 27 September 2021 |
| Mexico City | Ministry of Works and Public Services and former collaborators of the abolished Authority of Public Space | Local government | 30 November 2019 |
| | Ministry of Mobility of Mexico City and former collaborators of the abolished Authority of Public Space | Local government | 29 November 2019 |
| | Metropolitan Autonomous University, Landscape program | Academic | 18 November 2019 |
| | The National University of Mexico | Academic | 26 November 2019 |
| | Taller de Inovacion Urbana | NGO | 28 November 2019 |
| | Barriopolis | NGO | 8 November 2019 |
| | Thorsten Architects | Private | 9 November 2019 |
| | Ministry of Agrarian, Territorial and Urban Development (SEDATU) | Federal government | 19 November 2019 |
| | Ministry of Agrarian, Territorial and Urban Development (SEDATU) | Federal government | 22 January 2020 |
| | Ministry of Agrarian, Territorial and Urban Development (SEDATU) | Federal government | 23 January 2020 |

**Table A3.** Literature review: database search and identified papers.

| Web of Science | | |
|---|---|---|
| **Date of search: 19 December 2022** | | |
| **Search words** | **Results** | **Titles of the results (selected papers for this study are in bold and underlined)** <br> **Repeated papers considered for this study are marked with this mark (\*)** |
| social participation + ICT + public space | 28 | 1. **Access to ICT in Poland and the Co-Creation of Urban Space in the Process of Modern Social Participation in a Smart City—A Case Study (article is repeated)** <br> 2. **(\*)ICT as a solution for the revitalization of public open space in private developments** <br> 3. Public Participation in Local Regeneration Programmes in Poland: Case Study of Olkusz <br> 4. ICTs as keys to the enhancement of public awareness about potential earth impacts <br> 5. Developing a Digital Co-Creation Assessment Methodology <br> 6. Digital Civic Participation in the Context of Modern Research <br> 7. Flânerie between Net and Place: Promises and Possibilities for Participation in Planning <br> 8. Citizen participation in urban development: shifting roles in transforming spaces of Budapest <br> 9. Online political participation, civic talk, and media multiplexity: how Taiwanese citizens express political opinions on the Web <br> 10. Effects of the built and social features of urban greenways on the outdoor activity of older adults <br> 11. Public visualization displays of citizen data: Design, impact and implications |

**Table A3.** *Cont.*

| | | |
|---|---|---|
| social participation + ICT + public space | 28 | 12. The Power of the Audience-Public: Interactive Radio in Africa<br>13. **Participation through place-based e-tools: A valuable resource for urban green infrastructure governance?**<br>14. Researching Personal Information on the Public Web: Methods and Ethics<br>15. SWOT-AHP analysis of the Korean satellite and space industry: Strategy recommendations for development<br>16. Smart cities and their domains—Future challenges for urban researchers?<br>17. The Civil City Framework for the Implementation of Nature-Based Smart Innovations: Right to a Healthy City Perspective<br>18. Combining the Digital, Social and Physical Layer to Create Age-Friendly Cities and Communities<br>19. Drawing as an experience. An advanced scenario for culture representation<br>20. Smart infrastructure by PPPs within the concept of smart cities to achieve sustainable development<br>21. Advancing values-based approaches to climate change adaptation: A case study from Australia<br>22. Power Structures and Human Development in Communities Under Indigenous Customary Law (usos y costumbres): Reality and Trends 2012–2018. The cases of Santa Ines del Monte and San Miguel Huautla, Oaxaca<br>23. What do people want in a smart city? Exploring stakeholder opinions, priorities and perceived barriers in a medium-sized city in the United States<br>24. New Paradigms for Commercial Benefits from India's Earth Observation Activities<br>25. Collective Intelligence in Polish-Ukrainian Internet Projects. Debate Models and Research Methods<br>26. Indicators of sustainability to assess aquaculture systems<br>27. Alternative to civil society governance: platform control over the third sector in China<br>28. The Mechanism of Household Waste Sorting Behaviour—A Study of Jiaxing, China |
| green infrastructure *OR* urban park *OR* green space + public participation *OR* participatory mapping *OR* citizen science + ICT *OR* PPGIS | 17 | 1. Mapping place values: 10 lessons from two decades of public participation GIS empirical research<br>2. **Understanding the use of urban green spaces from user-generated geographic information**<br>3. **Geolocated social media as a rapid indicator of park visitation and equitable park access**<br>4. Using fuzzy cognitive mapping as a participatory approach to analyse change, preferred states, and perceived resilience of social-ecological systems<br>5. Assessing, mapping, and quantifying cultural ecosystem services at community level<br>6. Outdoor Activity Participation Improves Adolescents' Mental Health and Well-Being during the COVID-19 Pandemic<br>7. Mapping ecosystem service capacity, flow and demand for landscape and urban planning: A case study in the Barcelona metropolitan region<br>8. A generalized adoption model for services: A cross-country comparison of mobile health (m-health)<br>9. Built environmental correlates of older adults' total physical activity and walking: a systematic review and meta-analysis<br>10. High spatial resolution three-dimensional mapping of vegetation spectral dynamics using computer vision<br>11. Promoting sustainable intensification in precision agriculture: review of decision support systems development and strategies<br>12. Mapping the impact of patient and public involvement on health and social care research: a systematic review<br>13. Conceptualizing energy democracy<br>14. Remaking Participation in Science and Democracy<br>15. Urban Governance and the Politics of Climate change<br>16. When is a forest a forest? Forest concepts and definitions in the era of forest and landscape restoration<br>17. Third data release of the Hyper Suprime-Cam Subaru Strategic Program |

**Table A3.** *Cont.*

| Scopus | | |
|---|---|---|
| **Date of the search: 19 December 2022** | | |
| **Search words** | **Results** | **Titles of the results (select papers are in bold and underlined)**<br>**Papers that repeatedly appear in the searches carried out are marked with this symbol (\*)** |
| social participation<br>+ ICT<br>+ public space | 52 | 1.  Relevance of Smart Economy in Smart Cities in Africa<br>2.  **Playful e-participation with Minecraft as a development tool for urban redesign: A case study**<br>3.  Methodological Approaches to Reflect on the Relationships Between People, Spaces, Technologies<br>4.  The Oxford Handbook of Technology and Music Education<br>5.  **Development of methods and practices of virtual reality as a tool for participatory urban planning: a case study of Vilnius City as an example for improving environmental, social and energy sustainability**<br>6.  Research and Innovation (Rii) Forum 2021<br>7.  Drawing as an experience. An advanced scenario for culture representation<br>8.  ICT support to reconstruct social meaning after a disaster<br>9.  Flânerie between Net and Place: Promises and Possibilities for Participation in Planning<br>10. **Management of Public Space Towards Liveable City: The Case of Hanoi, and Lessons from Singapore**<br>11. Urbanization and smart cities<br>12. What do people want in a smart city? Exploring stakeholder opinions, priorities and perceived barriers in a medium-sized city in the United States<br>13. International Conference on Information Systems, ICIS 2013, Volume 3<br>14. Information and Communication Technologies (ICTs) as keys to the enhancement of public awareness about potential earth impacts<br>15. IFIP WG 3.4 International Conference on Open and Social Technologies, OST 2012<br>16. Technology, education and access: A 'fair go' for people with disabilities<br>17. Designing web 2.0 tools for online public consultation<br>18. New Paradigms for Commercial Benefits from India's Earth Observation Activities<br>19. Developing a digital co-creation assessment methodology<br>20. Augmented reality as a tool for open science platform by research collaboration in virtual teams<br>21. Digital media and political citizenship: Facebook and politics in South Africa<br>22. **Access to ICT in Poland and the co-creation of Urban space in the process of modern social participation in a smart city-a case study**<br>23. Innovation, technologies, participation: new paradigms towards a 2.0 citizenship<br>24. International Conference on Information Systems, ICIS 2013, Volume 2<br>25. Local eGovernment in the city of Casey: Political barriers to citizen engagement<br>26. International Conference on Information Systems, ICIS 2013, Volume 4<br>27. Social Implications of New Mediated Spaces: The Need for a Rethought Design Approach<br>28. Smart infrastructure by (PPPs) within the concept of smart cities to achieve sustainable development<br>29. Tracing two faces of extended visibility: a bibliometric analysis of transparency discussions in social sciences<br>30. **Digital tools for capturing user's needs on urban open spaces: drawing lessons from a cyberparks project**<br>31. Transforming government agencies' approach to e-participation through efficient exploitation of social media<br>32. **"Gamification" for Teaching Collaborative Urban Design and Citizen Participation**<br>33. Using virtual accessibility and physical accessibility as joint predictors of activity-travel behavior<br>34. Inequality in ICT access and its influence on media competency<br>35. Assessing public participation for the United Arab Emirates eGovernment<br>36. Technology for the independent living of people with activity limitations<br>37. Smart city in urban design<br>38. Public participation in local regeneration programmes in Poland: Case study of Olkusz<br>39. Sentiment and Visual Analysis: A Case Study of E-Participation to Give Value to Territorial Instances |

**Table A3.** *Cont.*

| | | |
|---|---|---|
| social participation + ICT + public space | 52 | 40. Online political participation, civic talk, and media multiplexity: How Taiwanese citizens express political opinions on the web |
| | | 41. **Collaborative platforms for social innovation projects. The Miramap case in Turin; [Piattaforme collaborative per progetti di innovazione sociale. Il caso Miramap a Torino]** |
| | | 42. From inclusive spaces to inclusionary texts: How e-participation can help overcome social exclusion |
| | | 43. The institutionalization of e-democracy: Challenges, risks and future directions in an Indian context |
| | | 44. Telephone booths as places of integrations: Information and communication technologies in the construction of networks and identities; [Los locutorios como espacios de integración: Las tecnologías de la información y la comunicación en la construcción de redes e identidades] |
| | | 45. Unpacking a smart city model: The hegemony of ecological and information paradigms in urban space |
| | | 46. International Conference on Information Systems, ICIS 2013, Volume 5 |
| | | 47. Innovative participatory evaluation processes: The case of the Ministry of Defence real estate assets in Italy |
| | | 48. Community, participation and virtual spaces: Design considerations for inclusivity |
| | | 49. **(*)ICT as a solution for the revitalization of public open space in private developments** |
| | | 50. The net, the public sphere and spaces of public deliberation; [La rete, la sfera pubblica e I luoghi della deliberazione pubblica] |
| green infrastructure *OR* urban park *OR* green space + public participation *OR* participatory mapping *OR* citizen science + ICT *OR* PPGIS | 18 | 1. **Coping With Crisis: Green Space Use in Helsinki Before and During the COVID-19 Pandemic** |
| | | 2. **Smart city in urban design** |
| | | 3. Public spaces as 'knowledgescapes': Understanding the relationship between the built environment and creative encounters at Dutch university campuses and science parks |
| | | 4. **(*)Understanding the use of urban green spaces from user-generated geographic information** |
| | | 5. **The pavilion of desires. Artistic co-creation for the improvement of public space [El pabellón de deseos. Co-creación y co-instalación artística para la mejora del espacio público]** |
| | | 6. **The added value of public participation GIS (PPGIS) for urban green infrastructure planning** |
| | | 7. Evaluation of ecosystem cultural services of urban protected areas based on public participation GIS (PPGIS): A case study of Gongqing Forest Park in Shanghai, China (in Mandarin) |
| | | 8. **An evaluation of participatory mapping methods to assess urban park benefits** |
| | | 9. More than A to B: Understanding and managing visitor spatial behaviour in urban forests using public participation GIS |
| | | 10. Smart city construction practices in BFSP |
| | | 11. **Public participatory mapping of cultural ecosystem services: Citizen perception and park management in the Parco Nord of Milan (Italy)** |
| | | 12. **Capturing residents' values for urban green space: Mapping, analysis and guidance for practice** |
| | | 13. **(*)Digital tools for capturing user's needs on urban open spaces: drawing lessons from cyberparks project** |
| | | 14. Using resident-based hazing programs to reduce human-coyote conflicts in urban environments |
| | | 15. The penetration of Information and Communications Technologies into public spaces: Some reflections from the Project CyberParks—COST TU 1306 [A agregação das Tecnologias de Informação e Comunicação ao espaço público urbano: reflexões em torno do Projeto CyberParks—COST TU 1306] |
| | | 16. Comparing conventional and PPGIS approaches in measuring equality of access to urban aquatic environments |
| | | 17. Smart cities and knowledge organisation |
| | | 18. **Using participatory GIS to measure physical activity and urban park benefits** |

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
