# Peer review of "The Use of ICTs to Support Social Participation in the Planning, Design and Maintenance of Public Spaces in Latin America"

_ijgi, doi:10.3390/ijgi12060237_

Round 1
Reviewer 1 Report
The article is well-organized and contains all of the components. The sections are well-developed. The authors did a good job in synthesizing the literature. The methodology used is clearly explained. However, I suggest minor spell check is required, for example:
In the line 15 - organisations, should be organizations.
In the line 49 - utilising, should be utilizing.
In the line 52 - analysed, should be analyzed.
In the line 162 - systematised, should be systematized.
Author Response
Reviewer 1 comments.
The article is well-organised and contains all of the components. The sections are well-developed. The authors did a good job in synthesising the literature. The methodology used is clearly explained. However, I suggest minor spell check is required, for example:
Response:
Dear reviewer, thank you for your kind words. We have made changes to our manuscript to provide consistency throughout our document. Our manuscript was developed using English (United Kingdom) as a base language, and the spelling of some words is different from American English. The minor spelling suggestions you are suggesting are based on the American English format. However, we checked the manuscript carefully to look for inconsistencies using British English, which is also the preferred language at our university.

Reviewer 2 Report
I can suggest the following references for further research:

Author Response
Dear Reviewer 2, Thank you for your kind words and the exhaustive list of literature recommendations. Below you will find how we have integrated your literature suggestions.
In chapter 2, we included a new paragraph that discusses studies that have emphasised the role of ICTs on which we used your suggested literature references. The paragraph on lines 102-111 reads as follows:
Current studies have emphasised how ICTs can be used to better align urban projects with residents' needs and aspirations. Examples of this are the use of interactive devices to increase participatory processes in the public space [1], or the exploration of affordable ICTs, based on the principles of digital democratic affordance, a concept that addresses how emergent digital tools could help to organise and represent the interest of social organisations [2]. Another approach involves operationalising different areas of knowledge and technologies to allow local residents to participate meaningfully in planning, designing or maintaining public spaces [3]–[5]. As long as digital platforms are not used as isolated solutions, they can become supportive tools that could enhance communication and interaction among residents and decision-makers to manage public spaces [7], [8].
The literature recommendation used is listed below:
Lee, D.; Lee, S. Investigating Media-User Interaction for Public Play Space in a Smart City. Appl.Sci. 2022, 12, 11882. https://doi.org/10.3390/app122311882
Deseriis M. 2021. Rethinking the digital democratic affordance and its impact on politicalrepresentation: Toward a new framework. New Media and Society 23(8): 2452-2473. DOI :10.1177/1461444820929678
Žlender V, Erjavec IŠ, Goličnik Marušić B. 2020. Digitally Supported co-creation within publicopen space development process: Experiences from the C3Places project and potential for futureurban practice. Planning Practice and Research 36(3): 247-26
Artopoulos, G., Arvanitidis, P., Suomalainen, S. (2019). Using ICT in the Management of PublicOpen Space as a Commons. In: , et al. CyberParks – The Interface Between People, Places andTechnology. Lecture Notes in Computer Science(), vol 11380. Springer, Cham.https://doi.org/10.1007/978-3-030-13417-4_14 https://link.springer.com/chapter/10.1007/978-3-030-13417-4_14#citeas

Reviewer 3 Report
Dear authors,
Thank you for a very interesting paper! In my opinion, it is of a good level and almost ready for publication. I only have some minor suggestions for further improvement:
· Section 3.1.1. on the PSM framework: provide a bit more details on what this framework entails / the core issues it identifies. Consider adding a visualization?
· In section 3.1.2. it remains a bit vague what “the quality of interaction between one or more stakeholders” entails, i.e. how do you determine this, what is “quality”? (line 152)
· For the literature review findings I would underline that this is based on a selection of articles and hence discusses academically documented ICT use in social participation, while in reality (outside of academic papers) the world might be more diverse.
· Why is there a specific table of ICT support focused on government officials and not on the other three actor groups?
· Line 471: the end of the quote, with the remark that “participatory results can be unpredictable” is actually quite funny.. isn’t the idea that by engaging locals a different source of knowledge is activated? If you would know the results beforehand then why would you bother, right?
· Before discussing the findings from the interviews I would suggest to add a short section explaining the Mexican context to the readers that are unfamiliar with this. In section 4.4 there are some sentences / info that would be useful earlier in the article, so the reader can better interpret/contextualize the findings. For example, line 511-513 on the digitisation of the government. And also in line 668-669.
· Lines 665-667 in the conclusion are a bit disconnected from the rest, while this is an important point that – I feel – deserves some more attention.
· For the conclusion, it might be nice to make a recommendation for including (more) ICT in planning education, something which you do shortly mention in line 47-50.
· Table A2 is missing 5 interviews?
And some very minor, more detailed, suggestions:
· There are some sentences that appear a bit random / disconnected from the rest of the paragraph. I suggest deleting them OR make sure to better integrate them in the running text:
o 1) “The paper also… in PDMPS” (line 64-65);
o 2) “Local governments … marginalised neighbourhoods” (line 116-117)
o 3) “However, it is … in PDMPS” (line 530-531)
· Line 22: specify who is lacking technical expertise
· Line 42: specify who is not accepting ICT
· Line 66: note that section 2 discusses the role of ICTs to support social participation in planning processes, not just ‘social participation in planning processes’ as is currently stated.
· Line 175: add a note that these four groups of actors are discussed in more detail further below.
· Check sentence flow of line 191-192-193.
· Line 292: presents > present
· Line 585: deleting “trying”?
· Line 585: specify “It”
· Line 665-666: isn’t and it’s > is not and it is
I’m looking forward to seeing the article published & thank the authors for their work!
Author Response
Dear Reviewer 3,
Thank you for your constructive suggestions. They helped us to increase the article's quality and legibility. Below you will find how we have integrated your suggestions into our manuscript. In the attached document, you can see the changes made according to your suggestions.

Reviewer 4 Report
Presented paper deals with, what the authors call, social participation in maintaining public spaces in Latin America. The paper is written in good English and it is well structured, but after reading the paper, I wondered why should this paper be published? What is the new information the paper presents?
Some of my comments regarding the paper are following:
Why do we (academics) have the urge to still invent some ridiculous abbreviations - such as PDMPS? Nobody uses this abbr. - on google scholar, there is exactly ONE paper using PDMPS and it is the paper from one of the authors of this one (see Sergio, Alvarado Vazquez, Ana Paula Rodriguez Müller, and Cesar Casiano Flores. "The use of ICTs in the planning and design process of public spaces, a co creation analysis." In WUR Conference, Date: 2022/01/24-2022/01/25, Location: Dortmund, pp. 13-14. Technische Universitât Dortmund, 2022.).
When speaking about local knowledge (line 33, for example), do you mean local spatial knowledge (LSK) - btw: abbreviation with several thousand references on google scholar...
Furthermore, while speaking about social participation - do you mean geoparticipation? Section 2 is very shallow, missing some key points/terms of the area.
Lines 108-9 digital inequality is not just about the level of access to ICT, but also about ICT literacy!
Line 121 - you are using PSM without explaining it, only later on line 130 you explain, what does it mean...
It is a bit unclear, how 15 papers you analysed were selected out of all 115.
I am also unsure, what are some tables (for example tab 3) good for - it is interesting to read it, but after reading it twice, I still have this "why"? in my head...
All in all, I do not think the paper is bad, but I am missing its point.
Author Response
Dear Reviewer 4,
Thank you for your constructive suggestions. They helped us to increase the article's quality and legibility. Below you will find how we have integrated your suggestions into our manuscript. In the attached document, you can observe the changes we made to our manuscript.
